# S100A8/A9 Enhances Immunomodulatory and Tissue-Repairing Properties of Human Amniotic Mesenchymal Stem Cells in Myocardial Ischemia-Reperfusion Injury

**DOI:** 10.3390/ijms222011175

**Published:** 2021-10-16

**Authors:** Tzu-Jou Chen, Yen-Ting Yeh, Fu-Shiang Peng, Ai-Hsien Li, Shinn-Chih Wu

**Affiliations:** 1Department of Animal Science and Technology, National Taiwan University, Taipei 106, Taiwan; zirouchen206@gmail.com (T.-J.C.); petere620@gmail.com (Y.-T.Y.); 2Cardiology Division of Cardiovascular Medical Center, Far Eastern Memorial Hospital, New Taipei City 220, Taiwan; las1012.tw@gmail.com; 3Department of Obstetrics and Gynecology, Far Eastern Memorial Hospital, New Taipei City 220, Taiwan; knox.p32@msa.hinet.net; 4Institute of Biotechnology, National Taiwan University, Taipei 106, Taiwan; 5Center for Biotechnology, National Taiwan University, Taipei 106, Taiwan

**Keywords:** human amniotic mesenchymal stem cells, myocardial infarction, cardiac remodeling, S100A8, S100A9, paracrine effects

## Abstract

Paracrine factors of human mesenchymal stem cells (hMSCs) have the potential of preventing adverse cardiac remodeling after myocardial infarction (MI). S100A8 and S100A9 are calcium-binding proteins playing essential roles in the regulation of inflammation and fibrous tissue formation, and they might modulate the paracrine effect of hMSCs. We isolated human amniotic mesenchymal stem cells (hAMSCs) and examined the changes in the expression level of regulatory genes of inflammation and fibrosis after hAMSCs were treated with S100A8/A9. The anti-inflammatory and anti-fibrotic effects of hAMSCs pretreated with S100A8/A9 were shown to be superior to those of hAMSCs without S100A8/A9 pretreatment in the cardiomyocyte hypoxia/reoxygenation experiment. We established a murine myocardial ischemia/reperfusion model to compare the therapeutic effects of the conditioned medium of hAMSCs with or without S100A8/A9 pretreatment. We found the hearts administered with a conditioned medium of hAMSCs with S100A8/A9 pretreatment had better left ventricular systolic function on day 7, 14, and 28 after MI. These results suggest S100A8/A9 enhances the paracrine therapeutic effects of hAMSCs in aspects of anti-inflammation, anti-fibrosis, and cardiac function preservation after MI.

## 1. Introduction

Approximately 17.9 million people worldwide die from cardiovascular diseases annually, and many of them suffer from myocardial infarction (MI) [1]. MI mostly results from obstruction of coronary arteries and causes myocardial tissue necrosis by ischemic injury. Subsequent inflammatory and fibrotic processes frequently lead to adverse cardiac remodeling and eventually heart failure. Mortality rates of acute MI have been reduced through development of the reperfusion strategy, however, this contributes to additional cell death caused by ischemia/reperfusion injury, and more effective therapies to prevent adverse cardiac remodeling are needed [2]. Cardiac fibrotic tissue might increase 10% to 35% and lead to irreversible cardiac function loss, rhythm irregularities, systolic disorders, and heart failure; hence, it is urgent to explore therapeutic options to prevent extensive inflammation and fibrosis after MI [3].

The message transmission and regulation of danger associated molecular patterns (DAMPs) released from the injured tissue play an essential role of immune regulation and wound healing [4]. DAMPs increase the permeability of endothelial cells, recruit neutrophils and monocytes to intensify inflammation, activate fibroblast proliferation, and accumulate extracellular matrix through interaction with cognate pattern recognition receptors. However, dysregulated responses of DAMPs promote progressive reactive fibrosis, leading to heart failure. Consequently, DAMPs are potential therapeutic targets in the development of post-MI heart failure [5]. In addition, DAMPs might play an essential role to regulate the characteristics of residential MSCs and myocardial repair and regeneration after cardiac injury [6]. Among them, S100 calcium-binding proteins have versatile functions, including cell proliferation, differentiation, migration, and energy metabolism by interacting with TLR2, TLR4, RAGE, and MAPK pathway after release from phagocytes. Previous research demonstrated that S100A9 blockade promotes adverse cardiac remodeling and aggravates ischemia/reperfusion (I/R) injury in MI [6].

Mesenchymal stem cells (MSCs) provide potential therapeutic options for various diseases due to unique regenerative and immunomodulatory abilities [7]. MSCs exhibit immunosuppressive properties via functional changing of immune cell populations in the wound area and inflammatory diseases [8,9]. Intramyocardial injection of MSCs reduced infarction size in prior animal experiments, but the risk of MSC therapy still needs further evaluation [10]. Timmers and colleagues demonstrated that treatment with conditioned medium of human MSCs reduced infarct area and significantly maintained post-MI cardiac contractile function in a porcine myocardial I/R model [11]. Human amniotic mesenchymal stem cells (hAMSCs) have many characteristics to be good MSC sources, including easy access, abundant sources, and no ethical disputes [12]. hAMSCs exhibited anti-apoptosis and pro-angiogenesis abilities by secreting extracellular vesicles, reduced infarction area, and improved cardiac function in previous animal MI models [13,14]. Some hAMSC-derived cytokines including TGF-β, HGF, PGE2, and IDO regulate relating molecular pathways in a dose-dependent manner, and Maleki and colleagues showed that intravenous injection of hAMSC-derived conditioned medium alleviated progression of heart failure by reducing fibrosis and increasing angiogenesis [15].

Prior research tried numerous cytokines or chemokines for pretreatment to enhance the therapeutic potential of MSCs. Basu and colleagues demonstrated that MSCs pretreated with S100A8/A9 accelerated wound healing and reduced scar tissue in a murine skin wound model. They showed that S100A8/A9 pretreatment improved the adaptive responses of MSCs, such as proteolysis, macrophage phagocytosis, and regulation of inflammation-related genes [16]. This study aims to investigate whether S100A8/A9 pretreatment enhances the paracrine therapeutic effects of hAMSCs in myocardial ischemia/reperfusion injury.

## 2. Results

This research included in vitro and in vivo experiments. Neonatal mouse cardiomyocytes (nCMs) and hAMSCs were isolated and cultured. Characteristics of hAMSCs were identified. The expression levels of interested genes, including S100 proteins, IL-1, IL-6, TNF-α, CRP, and IFN-γ, between control nCMs and hypoxia/reoxygenation (H/R) nCMs were compared. We also compared the expression levels of immunomodulatory and tissue repair-related genes, including TLR, CCR7, IL-10, GDF9, SPARCKL, MMP, and TIMP, between hAMSCs with and without S100A8/A9 pretreatment. We used the H/R model of nCMs to investigate in vitro anti-inflammatory and cardioprotective effects of conditioned medium derived from hAMSCs with or without S100A8/A9 pretreatment. At last, we investigated in vivo therapeutic effects of conditioned medium derived from hAMSCs with or without S100A8/A9 pretreatment in the murine ischemia/reperfusion (I/R) model. Serial measurements of left ventricular ejection fractions and fractional shortening were performed with echocardiography on day 1, 7, 14, and 28 after I/R injury. Histological transverse sections of left ventricles and quantification of fibrotic tissue areas with Masson’s trichrome staining were done. The study flowchart is shown in Figure 1.

### 2.1. Identification and Characterization of hAMSCs

After isolation and adequate culture, hAMSCs at passage 7 were spindle-shaped in morphology and developed a spiral growth pattern as seen in Figure 2A. The cell surface antigen analysis through flow cytometry is shown in Figure 2C. hAMSCs expressed mesenchymal stem cell-related surface markers and were negative with hematopoietic surface markers. The cell proliferation curve of P7 hAMSCs is shown in Figure 2B. We confirmed the ability of mesenchymal trilineage differentiation of hAMSCs in Figure 2D.

### 2.2. S100A8/A9 Pretreatment of hAMSCs

We treated hAMSCs with S100A8/A9 for 24 h and checked the change of expression levels of interested genes with qPCR. The results showed that S100A8/A9 treatment significantly altered the expression level of toll-like receptors (TLRs), tissue repair-related genes (SPARC, MMP2, MMP9, and TIMP), and immunomodulatory genes (CCR7 and IL10) in hAMSCs. The qPCR results are shown in Figure 3.

### 2.3. H/R Injury of nCMs In Vitro

To establish the H/R model in vitro, we optimized the procedure of H/R injury induction in nCMs, which is shown in Appendix A. Then we checked and compared the expression levels of interested genes in nCMs and H/R-injured nCMs. The qPCR results are shown in Figure 4.

### 2.4. S100A8/A9 Pretreatment Enhanced the Anti-Inflammatory Effects of hAMSC-Derived Conditioned Medium in the H/R Model

As shown in Figure 5, the expression levels of S100A1, S100A8, S100A9, CRP, IL-1α, IL-6, TNF-α, and IFN-γ in H/R-injured nCMs were significantly lowered by hAMSC-derived conditioned medium, and S100A8/A9 pretreatment further enhanced the immunomodulatory effects.

### 2.5. S100A8/A9 Pretreatment Enhanced the Therapeutic Effects of hAMSC-Derived Conditioned Medium in the Murine I/R Model

The left ventricular ejection fractions (LVEFs) and fractional shortening (FS) indexes of each group on day 1, 7, 14, and 28 after the I/R injury were measured by the high frequency ultrasound system as shown in Figure 6. The LVEFs and FS of the groups receiving hAMSC-derived conditioned medium treatment were significantly better than those of the I/R group on day 1, 7, 14, and 28. The group receiving hAMSC-derived conditioned medium with S100A8/A9 pretreatment (pre-hAMSC) further outperformed the one without S100A8/A9 pretreatment (non-hAMSC) on day 7, 14, and 28 after the I/R injury.

On day 28 after the I/R injury, the mice were anesthetized and sacrificed by cervical dislocation. The heart was quickly taken out and fixed with formalin. Cardiac fibrotic area was detected and measured after we stained the tissue sections of each group with Masson’s trichrome. Non-hAMSC and pre-hAMSC groups had less fibrotic area compared to the I/R group.

There were no significant differences in LV systolic function between pre-hAMSC and non-hAMSC groups on day 1 after the I/R treatment; but, thereafter, the differences increased, became significant on day 7, and persisted through day 28 after the I/R injury. The result of the in vivo experiments is reasonable and consistent with our in vitro data. The lack of differences in LV function on day 1 are not surprising since S100A8/A9 pretreatment did not provide an additional protective effect on acute cardiomyocyte death (Figure 5A). However, the pre-hAMSC group exhibited better maintenance of post-MI LV systolic function on a long-term basis, which presumably originates from the enhancement of anti-fibrosis and immunomodulatory properties with S100A8/A9 pretreatment demonstrated in the in vitro experiments (Figure 3 and Figure 5C).

## 3. Discussion

MSCs provide therapeutic potential for cardiovascular diseases due to paracrine and immunomodulatory properties [9]. MSC-derived conditioned medium has been investigated to attenuate heart failure in the animal model [15]. The production of extracellular vesicles from the stem and progenitor cells have been demonstrated to possess regenerative properties, and may provide a therapeutic opportunity post-MI [17]. However, when it comes to clinical translation, MSC-derived conditioned medium is still facing the problem of unstable quality, which originates from the heterogenous nature of MSCs and is largely influenced by different cultural conditions [18,19]. Therefore, previous studies have tried various pretreatments to enhance the therapeutic performance of MSCs [20].

Basu and colleagues showed that MSCs have unique characteristics after exposure to S100A8/A9 molecules [16]. In our research, hAMSCs were pretreated with S100A8/A9, which significantly improved the gene expressions of tissue repair (SPARC, MMP2, MMP9, and TIMP), inflammatory regulation (CCR7 and IL10), and TLRs. S100A8/A9 activates TLRs [21], stimulates the innate immune system in various diseases, and enhances the high expression of the MSC recruitment gene CCR7, which helps to gather autologous MSCs in adjacent wound sites [22]. In addition, the proteolytic balance of damaged tissue played an essential role in regulating the wound debridement and inflammation responses. The proteolytic regulation may be influenced by the overexpression of these matrix metalloproteinases and the decreased expression of the inhibitors, but the mechanism and pathway of these molecules have not yet been defined [23,24,25]. SPARC is also involved in the formation of cell matrix [26], and IL-10 has the ability to recruit regulatory T cells with anti-inflammatory properties. According to recent studies, MSCs with a high expression level of IL-10 increase stem cell retention in the tissue, promote tissue repair, protect muscles from physical damage, and thereby alleviate muscle dysfunction [27].

There are numerous potential applications of MSC-derived conditioned medium in regenerative medicine, but there still exists many problems to be solved [18]. One of the most important problems is instability of paracrine components in MSC-derived conditioned medium, which will largely influence the therapeutic effect. In our research, the experimental conditioned medium did not contain additional serum to minimize the interference of other growth factors. In previous studies, it was pointed out that this may significantly affect the concentration of growth factors such as HGF, FGF2, VEGF, and PEDF released by MSCs, thereby affecting the characteristics of MSCs [18]. The components of MSC-derived conditioned medium are different among various cell sources and greatly impact on the therapeutic potential. Moreover, accumulation of paracrine factors during cell proliferation is a dynamic process instead of a steady one, greatly influenced by different cultural conditions and protocols. Therefore, in order to optimize the therapeutic effect, the procedure of collecting MSC-derived conditioned medium is to be standardized for various types of MSCs in the future. In our opinion, in addition to appropriate cell density and enough duration of conditioned medium collection, adequate pretreatment that has been proved to enhance therapeutic potential of MSCs, e.g., hypoxia, cytokines, or certain chemical compounds, should be applied according to different target diseases. S100A8/A9 was proved to be a potential candidate pretreatment agent to enhance the therapeutic effect of MSC-derived conditioned medium in MI.

The timing of treatment with MSC-derived conditioned medium might play an important role. However, models mimicking the clinical situation more closely should be used in a clinically relevant experimental study according to prior consensus [28]. Therefore, we administered the conditioned medium treatment at the moment when cardioprotective therapies are most likely given in clinical practice, which is late ischemia and the beginning of coronary reperfusion. In summary, this study demonstrated that the conditioned medium of S100A8/A9-pretreated hAMSCs improved cardiac systolic function and decreased myocardial fibrosis after I/R injury. In addition, S100A8/A9 pretreatment enhanced the immunomodulatory and tissue-repairing properties of hAMSCs by changing relevant gene expressions and, thus, might contribute to optimization of the procedures of cell-based therapy in clinical applications.

## 4. Materials & Methods

### 4.1. Animals

The experimental animal projects were qualified by the examination of the Institutional Animal Care and Use Committee of National Taiwan University (IACUC, 109-00016). C57BL/6 male mice (12 weeks old) were purchased from the National Taiwan University of Medicine Laboratory Animal Center and bred in-house. Mice were fed a normal laboratory diet with water.

### 4.2. Preparation of hAMSCs

Human amniotic mesenchymal stem cells were obtained from Far Eastern Memorial Hospital, with the isolation and culture methods as previously described [29]. The normal cultivate condition uses Dulbecco’s Modified Eagle Medium (Gibco) with 10% fetal bovine serum (Merk, Kansas, MO, USA) and 1% Antibiotic-Antimycotic (Thermo, Waltham, MA, USA) at 37 °C and 5% carbon dioxide in a humidified atmosphere. The adherent cells were passaged with 1× TrypLE (Gibco, Amarillo, TX, USA) containing 1nM EDTA (Invitrogen, Waltham, MA, USA).

### 4.3. Preparation of nCMs

Neonatal mouse cardiomyocytes (nCMs) were isolated and cultured from C57BL/6 mice within 24 h of birth, digested in 10× TrypLE, and centrifuged. Then, the cell suspensions were plated twice, once for 1.5 h to remove non-myocytes and plated on 6-well plate coated with 0.5% gelatin (Sigma, St. Louis, MO, USA) at a density of 1250 cells/mm^2^. 5-bromo-2′-deoxyuridine (Sigma) was added to medium and experiments were initiated 24 h later [30].

### 4.4. H/R Injury Induction of nCMs

Hypoxia was achieved by incubating the nCMs in a hypoxia chamber in the absence of serum and glucose with an atmosphere of 1% O_2_, 5% CO_2_ at 37 °C for 24 h. Cells were then reoxygenated under normoxic conditions in a 95% N_2_, 5% CO_2_ humidified atmosphere at 37 °C for 6 h in Dulbecco’s Modified Eagle Medium containing glucose. Cells incubated at 37 °C under normoxic conditions were used as normoxic control cells [31].

### 4.5. S100A8/A9 Pretreatment of hAMSCs

The conditioned medium was prepared containing Human S100A8 and S100A9 Heterodimer Protein (Sino Biologycal, Beijing, China) at a concentration of 5μg/mL in Dulbecco’s Modified Eagle Medium without additional serum and other growth factors. Human AMSCs were then cultivated with above conditioned medium at 37 °C, 5% CO_2_, 20% O_2_ for 24 h.

### 4.6. Cell Proliferation Test

The Cell Counting Kit-8 (CCK-8, Dojindo) colorimetric method was used to measure the number of viable cells on day 0, 2, 4, 6, 8, and 10 after passage to the seventh generation of hAMSC in a normal cultural condition. We added 10 μL CCK-8 to a 96-well plate containing 100 μL culture medium and hAMSCs during the measurement. After two hours of co-cultivation in the incubator, we transferred the culture medium containing CCK-8 to the new one. The results were analyzed with an ELISA reader at a wavelength of 450 nm. The calculation method used the standard curve to convert the number of cells at each day.

### 4.7. Mesenchymal Trilineage Differentiation Tests of hAMSCs

The hAMSCs were cultured using MesenCult™ Adipogenic Differentiation Kit (STEMCELL TECHNOLOGIES, Inc, Canada) and MesenCult™ Osteogenic Differentiation Kit (STEMCELL TECHNOLOGIES, Inc., Canada) to induce differentiation. Cell pellets (containing 2 × 10^6^) were cultured for 21 days in MesenCult™-ACF Chondrogenic Differentiation Kit (STEMCELL TECHNOLOGIES, Inc, Canada) to induce differentiation. Control cells for all treatments were cultured under normal conditions as previously described. After the induced differentiation stage, the cells were stained with alizarin red to assess osteogenic differentiation, Oil Red O to assess adipogenic differentiation, and Alcian blue for chondrogenic differentiation. Chondrogenic pellets were paraffin-embedded and 3 μm sections were taken, which were then rehydrated and further stained with 1% Alcian blue solution.

### 4.8. Flow Cytometry

hMSCs were identified via flow cytometry with FITC-conjugated antibodies against FITC-anti CD90 (BD Bioscience, Franklin Lakes, NJ, USA, 1:100), FITC-anti CD45 (eBioscience, San Diego, CA, USA, 1:100), FITC-anti CD34 (eBioscience, 1:100), FITC-anti CD11b (eBioscience, 1:100), FITC-anti CD105 (eBioscience, 1:100), FITC-anti CD73 (eBioscience, 1:100), FITC-anti CD14 (eBioscience, 1:100), and FITC-anti mouse IgG as negative control.

### 4.9. RNA Isolation

After collecting the nCMs or the hAMSCs using 500 μL of GENEzolTM Reagent (Geneaid) and 50 μL 1-bromo-3-chloropropane (BCP, MRC), we homogenized the cell pellet and reacted at room temperature for 15 min, then centrifuged at 1600× *g* at 4 degrees Celsius for 15 min. The upper aqueous layer was taken and mixed with isopropanol (Merck, MA, USA) in the same volume after centrifugation at 1600× *g* at 4 °C for 10 min. The supernatant needed to be removed, the RNA at the bottom of the tube rinsed with 75% alcohol and centrifuged at 1600× *g* at 4 °C for 1 min, and this wash step repeated three times. After the alcohol evaporated, it was re-dissolved with 20 to 30 μL of ddH_2_O. Finally, the OD value of RNA was measured with Nanodrop and stored at 80 degrees Celsius.

### 4.10. Quantitative Real-Time PCR

Total RNA was reversely transcribed to cDNA by using DNaseⅠ amplification Grade kit (Invitrogen, Waltham, MA, USA) and then using High-Capacity RNA-to-cDNA Kit. Final complementary DNA (cDNA) product was stored at −20 degrees Celsius. Real-time PCR was performed using SimplyGreen qPCR Master Mix (Simply). GAPDH was used as the internal control. The used primers are shown in Appendix A.

### 4.11. Preparation of S1008/A9-Pretreated hAMSC-Derived Conditioned Medium

We seeded approximately 1 × 10^5^ of hAMSCs into the 10 cm culture dish and replaced the general culture medium with 10 mL conditioned medium containing Human S100A8 & S100A9 Heterodimer Protein (as previously described) when the cell density was up to 1 × 10^6^. After pretreatment for 24 h, we changed the conditioned medium into the serum-free medium and kept it cultivated for 48 h. After that, we collected the supernatant and concentrated 25-fold of the hAMSC derived conditioned medium through Amicon Ultra-15 Centrifugal Filter Devices (Millipore, Darmstadt, Germany), the final volume was about 0.5 mL of each 10 cm plate.

### 4.12. Murine Myocardial I/R Model

The murine myocardial I/R model was established using C57BL/6 male mice (*n* = 36), which were anesthetized by intraperitoneal injection of pentobarbital sodium (60 mg/kg) and ventilated via tracheal intubations connected to a rodent ventilator. The chest was opened by a horizontal incision at the third intercostal space. Myocardial ischemia was produced by left anterior descending coronary artery ligation using 6-0 silk suture. Successful performance of myocardial infarction model was confirmed by observing a pale region below the ligation areas. After 30 min of myocardial ischemia, 200 μL of the conditioned medium for each treatment group was intravenously infused through tail veins before the ligation was released to allow reperfusion. Sham-operated mice (*n* = 15) undergoing the same surgical procedures, except coronary artery ligation, were used as control.

### 4.13. Echocardiography

Mice were anesthetized as described above. Transthoracic echocardiography was performed with High-Frequency Ultrasound (Prospect 3.0) equipped with a 35 MHz phased array. Recordings were obtained from all mice at day 1, 7, 14, and 28 after the I/R surgery. Hearts were imaged two-dimensionally in the parasternal long- and short-axis views, through which the M-mode cursor was positioned perpendicular to the left ventricular septum and posterior wall. All measurements were performed by a trained technician and were averaged for at least three consecutive cardiac cycles.

### 4.14. Histological Staining and Quantification Analysis

Hearts were perfused with 4% formaldehyde and cut into two transverse slices parallel to the atrioventricular ring. Each slice was fixed with 4% buffered formalin, embedded in paraffin, and sectioned into 5-µm slices. Serial sections were stained with either hematoxylin and eosin to detect cellular infiltration or with Masson’s trichrome stain (Sigma) to detect fibrosis and assess its area. All slides were digitally photographed and analyzed with ImageJ.

### 4.15. Statistical Analysis

All the data were expressed as mean ± SD. Statistical comparison of the data was performed using the student *t*-test for comparison between two groups, and repeated measurement ANOVA for comparison among groups followed by Tuckey’s multiple comparison test. *p* < 0.05 was considered statistically significant.

## 5. Conclusions

This study demonstrated intravenous administration of the conditioned medium of S100A8/A9-pretreated hAMSCs improved left ventricular function and decreased myocardial fibrosis after I/R injury. S100A8/A9 pretreatment enhanced the paracrine immunomodulatory and tissue-repairing properties of hAMSCs by changing relevant gene expressions. It might contribute to optimization of the procedures of cell-based therapy in clinical applications, and further research is warranted.

## Figures and Tables

**Figure 1 ijms-22-11175-f001:**
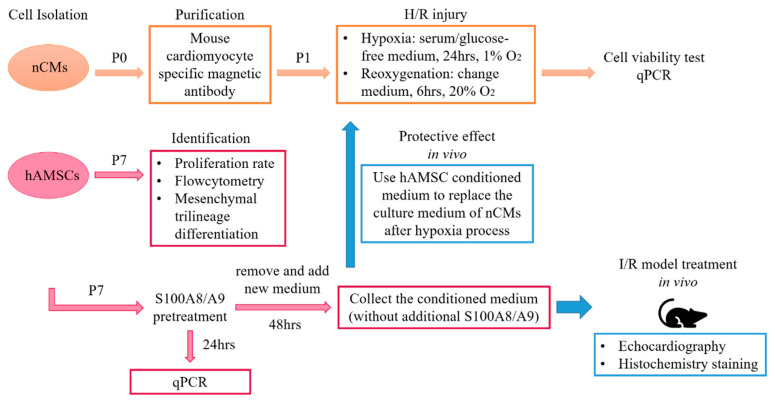
The study flowchart.

**Figure 2 ijms-22-11175-f002:**
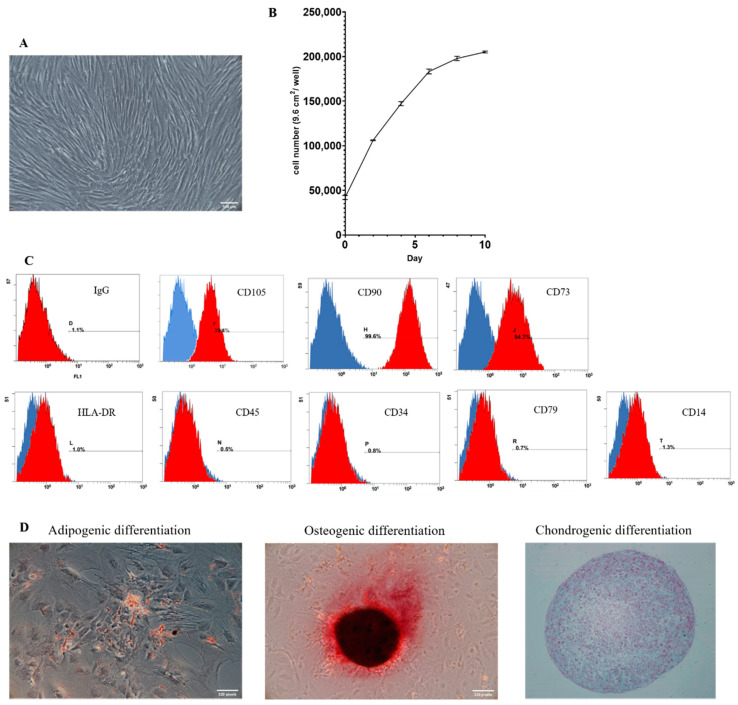
The characteristics of hAMSCs. (**A**) Morphology of hAMSC at passage 7 under normal cultural condition. The morphology of hAMSCs show accurate spindle-shape under microscopic observation (Bar = 100 μm). (**B**) Cell proliferation test of hAMSC at passage 7. Measured the cell proliferation at day 0, 2, 4, 6, 8, and 10 after seeded approximately 4 × 10^4^ of hAMSCs in 6-well plate through CCK-8 kit and ELISA determination (OD: 450 nm). (**C**) Flow cytometry analysis of hAMSCs surface marker expression. Flow cytometry results of hAMSCs at passage 7 represented the CD105, CD90, and CD73 positive. (**D**) Representative microscopic images of Alizarin Red S staining and Oil Red O staining to confirm osteogenic differentiation and adipogenic differentiation of plastic adherent hAMSCs in passage 7 after 14 days induced. Chondrogenic differentiation tests were confirmed by the formation of dense cell pellets and Alcian blue staining after 21 days induced. (Bar = 100μm (**A**,**B**); Bar = 200 μm (**C**)).

**Figure 3 ijms-22-11175-f003:**
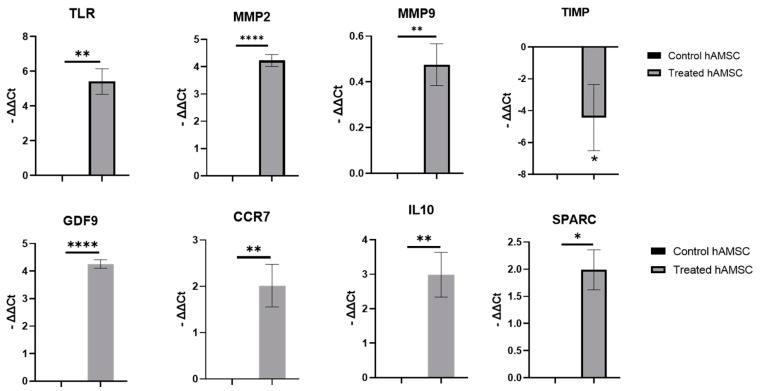
The comparison of expression levels of interested gene in hAMSCs with or without S100A8/A9 treatment. The significance among the groups was calculated using *t* test. * *p* < 0.05, ** *p* < 0.01, **** *p* < 0.0001.

**Figure 4 ijms-22-11175-f004:**
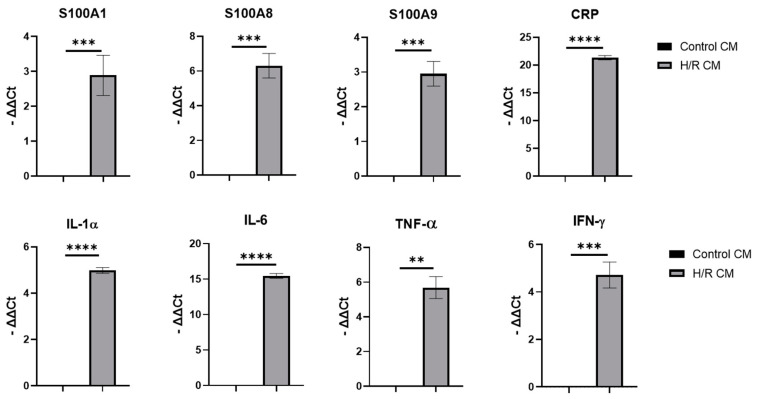
Analysis of the gene expressions in control cardiomyocyte group (Control CM) and H/R cardiomyocyte group (H/R CM) through qPCR. The significance among the groups was calculated using *t* test. ** *p* < 0.01, *** *p* < 0.001, **** *p* < 0.0001.

**Figure 5 ijms-22-11175-f005:**
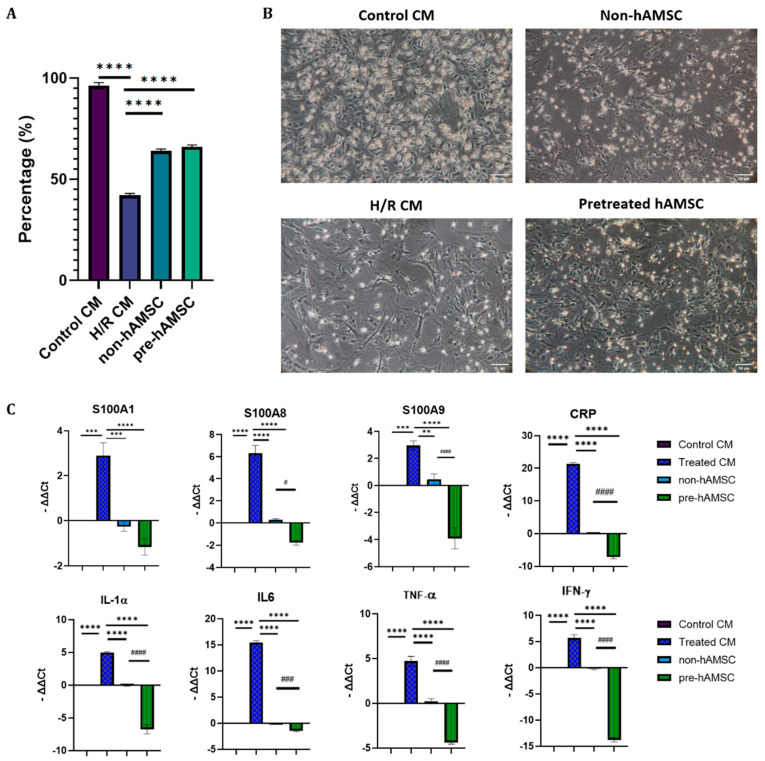
The immunomodulatory and cardioprotective effects of hAMSC-derived condition medium in the H/R model. (**A**) Cell viability tests of control, H/R CM, non-pretreated hAMSC (non-hAMSC), and pretreated hAMSC (pre-hAMSC) groups through trypan blue staining. (**B**) Microscopic images of each group after different treatment. (**C**) Gene expression levels in each group through qPCR. The significance among the groups was calculated using one-way ANOVA. * each group compared with H/R group. ** *p* < 0.01, *** *p* < 0.001, **** *p* < 0.0001. ^#^ pre-hAMSC group compared with non-hAMSC group. ^#^ *p* < 0.05, ^###^ *p* < 0.001, ^####^
*p* < 0.0001.

**Figure 6 ijms-22-11175-f006:**
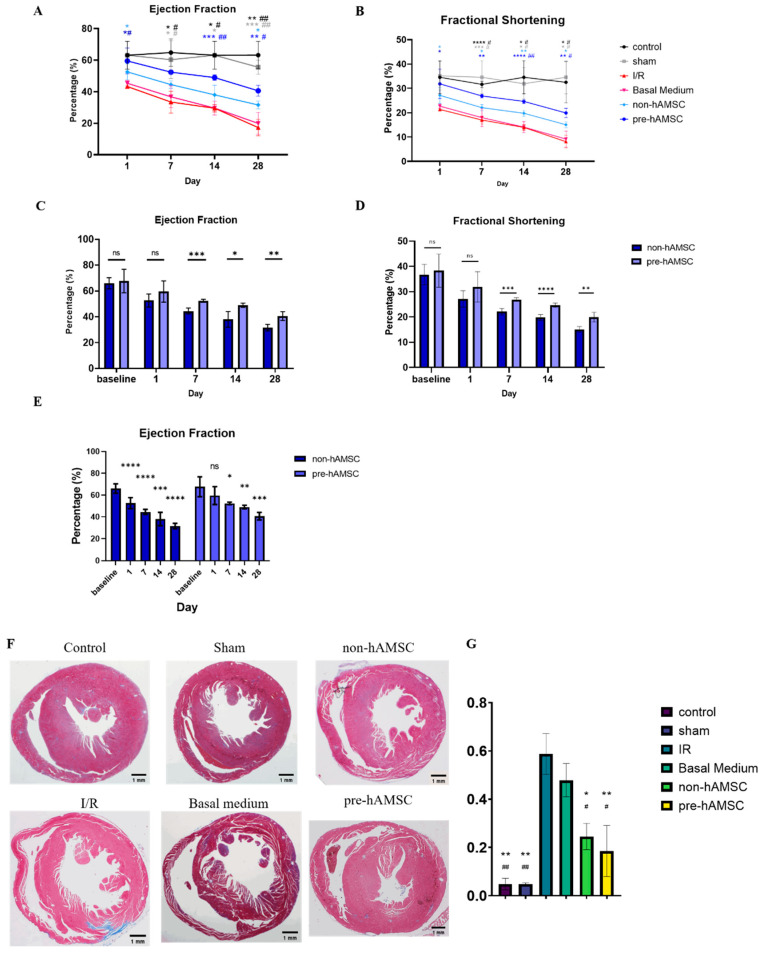
Cardiac function and fibrosis in each group during the 4-week period after the I/R injury. LVEFs (**A**) and FS (**B**) of control, sham, I/R, basal medium, non-hAMSC, and pre-hAMSC groups. (**C**–**E**) Representative bar graphs of left ventricular function in non-hAMSC and pre-hAMSC groups. (**F**) Masson’s Trichrome staining of histological transverse sections of hearts in each group 28 days after the I/R injury (Bar = 1 mm). (**G**) Quantification of heart fibrosis area in each group at 28 days after I/R surgery via Masson’s Trichrome staining. * Comparison with I/R group. * *p* < 0.05, ** *p* < 0.01, *** *p* < 0.001, **** *p* < 0.0001. # Comparison with Basal Medium group. ^#^ *p* < 0.05, ^##^ *p* < 0.01.

## Data Availability

Datasets analyzed during the present study are available from the corresponding author upon reasonable request.

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
