# Peer review of "S100A8/A9 Enhances Immunomodulatory and Tissue-Repairing Properties of Human Amniotic Mesenchymal Stem Cells in Myocardial Ischemia-Reperfusion Injury"

_ijms, 2021, doi:10.3390/ijms222011175_

Round 1

Reviewer 1 Report

This work by Chen et al. is a well-presented overview of the effects of S100A8/A9-treated human amniotic mesenchymal stem cells (hAMSCs) on neonatal mouse cardiomyocytes (nCMs) subjected to hypoxia/reoxygenation (H/R) in vitro, and murine myocardial ischemia/reperfusion (MI) in vivo. The setup, results and conclusions are very clear and to-the-point. I only have some minor comments:

  • Despite the effects on mRNA expression and the results in the MI model, survival of nCMs was not different after addition of non- vs. pre-treated hAMSCs (Fig. 5A). Perhaps this is due to the timing of reperfusion? A difference might be observed sooner or later post-reperfusion. It would be more interesting to observe a survival curve with a measurement each hour starting from the moment of medium replacement (after hypoxia).
  • In this context, it would be interesting to assess cell death in the in vivo MI model (e.g. by TUNEL staining).
  • Did all the animals survive the I/R treatment, and if not, was there any effect on survival?

Reviewer 2 Report

The manuscript titled “S100A8/A9 Enhances Immunomodulatory and Tissue-Repairing Properties of Human Amniotic Mesenchymal Stem Cells in Myocardial Ischemia-Reperfusion Injury” by Chen et al is a well-written research paper mainly investigated how hAMSCs response to S100A8/A9 treatment and how it contributes to the treatment of myocardial infarction (MI). Compared to untreated hAMSCs, the results showed that those pretreated with S100A8/A9 molecules demonstrated enhanced tissue repair ability, inflammatory regulations, and expressions of toll-like receptors (TLRs). This finding is meaningful. Data has been clearly presented and appropriately discussed. A minor revision is recommended.

  1. Small language problems especially some inappropriate article usage.
  2. Page 3, descriptions for Fig. 1A-D are not right. These should be Fig. 2A-D instead.
  3. Fig. 5C, it is recommended to use either different colors or patterns for different group on the bar chart.
  4. Fig. 3, why did the control group of SPARC expression show negative value? What house-keeping gene is used here? GAPDH?
  5. Does the authors have any opinions on how to improve the quality of MSC-derived conditioned medium or standardize the procedures of collecting the medium? It is recommended to include some suggestions in the discussion part.
  6. The in vivo data has not been well discussed. There is not enough mention on the differences among the 6 different groups.

Round 2

Reviewer 1 Report

I have no further comments.